# Associations of Adipocytokines and Early Renal Dysfunction in Young People on the Background of Dyslipidemia

**DOI:** 10.3390/jpm13081238

**Published:** 2023-08-09

**Authors:** Evgeniia V. Garbuzova, Alyona D. Khudiakova, Lilia V. Shcherbakova, Elena V. Kashtanova, Yana V. Polonskaya, Ekaterina M. Stakhneva, Yulia I. Ragino

**Affiliations:** Research Institute of Internal and Preventive Medicine–Branch of the Institute of Cytology and Genetics, Siberian Branch of Russian Academy of Sciences (IIPM—Branch of IC&G SB RAS), B. Bogatkova Str., 175/1, 630089 Novosibirsk, Russia; stryukova.j@mail.ru (E.V.G.); alene.elene@gmail.com (A.D.K.); 9584792@mail.ru (L.V.S.); elekastanova@yandex.ru (E.V.K.); yana-polonskaya@yandex.ru (Y.V.P.); stahneva@yandex.ru (E.M.S.)

**Keywords:** glomerular filtration rate, kidney dysfunction, adipokines, young population, dyslipidemia

## Abstract

Background: There are data supporting the idea that atherogenic dyslipidemia is a risk factor for CKD and reduced GFR. The aim was to evaluate the associations between adipocytokines and early renal dysfunction in young people with dyslipidemia. Materials and methods: A population study was conducted in IIPM—Branch of IC&G SB RAS, in 2013–2017. Furthermore, 1033 people were included in the study (469 men (45.4%) and 564 women (54.6%)). The study included blood sampling, anthropometric data, and adipokines by multiplex analysis. Results: Among people with reduced kidney function and DLP, men were 3.1 times more common than without DLP, women smoked 2 times less often, arterial hypertension was 7.8 times more common, and abdominal obesity was 2.7 times more common (and women with DLP were 3 times more likely than those without DLP). An increase in the level of resistin by 1 mcg/mL was associated with an increased chance of having renal dysfunction by 0.2%. An increase in the level of GIP was associated with an increased chance of having renal dysfunction by 1.1%. Conclusions: In young people with dyslipidemia, regardless of the presence of abdominal obesity, resistin and GIP are associated with the presence of renal dysfunction.

## 1. Introduction

Chronic kidney disease (CKD) has been recognized as an important social problem worldwide. The prevalence of CKD is approximately 13.4% worldwide [1], and, according to the ESSE-RF-2 study, 29.0% of people in Russia are characterized by an initial decrease in GFR of less than 90 mL/min/1.73 m^2^ and 1.6% have a reduced GFR of less than 60 mL/min/1.73 m^2^ [2]. It has been demonstrated that CKD is the main risk factor for mortality from all causes, and a moderate increase in serum creatinine levels was associated with an increase in mortality from any cause [3,4,5]. The progression of kidney damage is closely related to systemic inflammation [6,7]. In addition, clinical studies have shown that elevated markers of inflammation, such as C-reactive protein (CRP) and interleukin-6 (IL-6), are associated with various complications of CKD, such as malnutrition, atherosclerosis, insulin and erythropoietin resistance, calcification of the coronary arteries, heart disease, anemia, and increased mortality [6,8,9].

There is epidemiological and clinical data supporting the idea that atherogenic dyslipidemia is a risk factor for the occurrence or progression of CKD and reduced GFR [10,11]. However, the relationship between serum levels of low-density lipoprotein cholesterol (LDL-C), as well as inflammatory markers secreted by visceral adipocytes, and cardiovascular diseases is more difficult to understand in the context of a decrease in GFR, especially at the very initial stages of renal dysfunction. Chronic inflammation is an additional factor that should be taken into account with the progression of a decrease in GFR, which may increase the risk of atherogenic dyslipidemia and cardiovascular events in the future. This study was aimed at evaluating the associations between adipocytokines and early renal dysfunction in persons aged 25–44 years in Novosibirsk with a background of dyslipidemia.

## 2. Materials and Methods

A population study of the Novosibirsk population aged 25–44 years was conducted in IIPM—Branch of IC&G SB RAS, in 2013–2017. The study was funded by a grant from the Russian Science Foundation No. 21-15-00022. This additional part of the study was approved by the local Ethics Committee Protocol No. 167 dated 26 November 2019 (biomarker study).

To collect the sample, the database of the Territorial Compulsory Health Insurance Fund for Persons aged 25 to 44 years was used for one of the districts of Novosibirsk, which belong to a typical population in terms of industry, social life, demography, transport structure, and migration level. A random number generator was used to create a representative sample of 2500 people. Since it is known that the most rigid response is observed among young age groups, methods of step-by-step stimulation were used, including invitations by mail, phone calls, and information messages in the media. The skinning survey included 1512 people, and the response rate was 60.5%. The study included 1033 people after excluding pregnant women and women on maternity leave, including 469 men (45.4%) and 564 women (54.6%). The average age of the study participants was 37.2 years [31.4; 41.8]. All signed an informed consent to the study and processing of personal data. 

A group of clinicians who had received training in standardized epidemiological methods of screening examinations carried out the screening. The survey program consisted of demographic and social information, a survey on drinking and smoking habits, a socioeconomic survey, a dietary survey, a history of chronic illnesses and medication use, the Rose Angina Questionnaire, anthropometry, a 2-fold measurement of blood pressure (BP), spirometry, an ECG recording with transcription in accordance with Minnesota code, and other measurements.

Heavy drinking was defined as follows: For men, consuming more than 4 drinks on any day or more than 14 drinks per week; for women, consuming more than 3 drinks on any day or more than 7 drinks per week. The average dosage for our population was as follows: For men, 1063 mL of beer, 41.3 mL of wine, and 77.77 mL of distilled spirits per week; for women, 289 mL of beer, 68 mL of wine, and 9 mL of distilled spirits per week.

After a 5-min rest, blood pressure was measured twice with a 2-min gap using an Omron M5-I automatic tonometer on the right hand while seated, with the average of the two readings recorded. Systolic blood pressure (SBP) 140 mmHg and/or diastolic blood pressure (DBP) 90 mmHg were documented as arterial hypertension (AH).

Body weight (kg) divided by height (m^2^) squared was the formula used to calculate the body mass index (BMI). A high BMI was defined as >25 kg/m^2^. With a centimeter tape placed horizontally in the midpoint between the sacral region of the ilium and the lower edge of the costal arch, waist circumference (WC) was measured. Men with a WC of 94 cm and women with a WC of 80 cm were found to have abdominal obesity [12,13].

Smokers were defined as those who smoked at least one cigarette each day.

Twelve hours after eating, a single blood sample was drawn from the ulnar vein on an empty stomach. On an automatic biochemical analyzer, the Konelab 30i (Finland, Vantaa), blood parameters for lipid profile, glucose, and creatinine were determined by the enzymatic approach using standard ThermoFisher reagents. The formula used to convert serum glucose into plasma glucose was plasma glucose (mmol/L) = −0.137 + 1.047 × serum glucose (mmol/L). Decreased blood levels of HDL-C were estimated to be 40 mg/dL for males and 50 mg/dL for women, while elevated blood levels of TG were estimated to be 150 mg [13]. Elevated blood levels of LDL-C were estimated to be 116 mg/dL. Deviations in at least one of the aforementioned parameters were used to define dyslipidemia.

The calculation of GFR was carried out according to the national recommendations of KDIGO (Kidney Disease: Improving Global Outcomes) 2012 according to the formula CKD-EPI (Chronic Kidney Disease Epidemiology Collaboration), taking into account race, gender, age, and serum creatinine. GFR ≥ 90 mL/min/1.73 m^2^ was considered high or optimal and <90 mL/min/1.73 m^2^ was considered low [14].

The renal dysfunction group included 239 people (23.1% of the total sample), who were divided into two subgroups: With renal dysfunction and the presence of dyslipidemia and with renal dysfunction without it (Figure 1). Persons without renal dysfunction (76.9% of the total sample) were also divided into two subgroups depending on the presence of dyslipidemia.

The levels of amylin, C-peptide, ghrelin, glucose-dependent insulinotropic polypeptide (GIP), glucagon, interleukin 6, insulin, leptin, monocytic chemotactic factor 1 (MCP-1), pancreatic polypeptide (PP), and tumor necrosis factor alpha were determined by multiplex analysis using the Human Metabolic Hormone V3 (MILLIPLEX) panel (TNF-α). The Human Adipokine Magnetic Bead Panel 1 was used to determine the levels of adiponectin, adipsin, lipocalin-2, plasminogen activator inhibitor type 1 (PAI-1), and resistin.

Using the SPSS software package (version 13.0), the collected findings were statistically processed. The sample size was calculated using the formula:$$n=\frac{t^{2}\times P\times Q}{\Delta^{2}}$$

According to calculations, the required sample size was 288 people.

The Kolmogorov–Smirnov criterion was used to verify the normal distribution. The data are presented for categorical variables in the form of absolute and relative values—n (%), and for continuous variables in the form of Me [25; 75], where Me is the median and 25 and 75 are the 1st and 3rd quartiles, respectively, due to the abnormal distribution of the majority of the studied indicators. Two independent samples were compared using the nonparametric Mann–Whitney U-test. The fractions were contrasted using Pearson’s chi-squared test. Multiple logistic regression analysis was used to evaluate the associations under the following conditions: Independence of residuals; absence of multicollinearity, or circumstances in which independent variables strongly correlate with one another (r > 0.7); linear dependence between each independent variable and the logarithm of the odds ratio (logarithmic coefficients); and the dependent variable being a dichotomous measure of the presence or absence of a decrease in GFR. The output of the multiple logistic regression analysis was displayed as an OR and its 95% confidence interval. The null hypothesis’ crucial significance level (*p*) was taken to be 0.05.

## 3. Results

At the first stage of our study, clinical and anamnestic data from people with and without renal dysfunction were analyzed. Individuals with reduced GFR were older than those with normal GFR and also had lower WC and SBP values. At the same time, the levels of TC, LDL-C, Non-HDL-C, and HDL-C were higher in the group with reduced GFR compared with the group with normal renal function (Figure 2 and Figure 3).

Individuals with reduced renal function and DLP had lower levels of WC, SBP, albumin, and HDL-C in the blood compared to individuals with normal renal function. In the group of individuals with reduced renal function without DLP, lower BMI, OT, SAD, DAD, albumin, and HDL cholesterol were observed compared to individuals with normal renal function. (Table 1). 

Among people with reduced kidney function and DLP, men were 3.1 times more common than without DLP, women smoked 2 times less often, arterial hypertension was 7.8 times more common, and abdominal obesity was 2.7 times more common (and women with DLP were 3 times more likely than without DLP). (Table 2).

The analysis of adipokine levels in the groups under investigation came next. The concentrations of adipsin, ghrelin, GIP, and PYY were 1.2 times greater in the group of individuals with renal dysfunction. The levels of IL-6, C-peptide, and TNFa in the group with reduced GFR were 1.4 times higher, and GLP1 and leptin were 1.6 times higher compared to the group with normal renal function. The value of lipocalin-2 was more than 2.3 times higher in individuals with a GFR of 90 mL/min/1.73 m^3^, the value of secretin was 1.5 times higher, the values of resistin were 7.7 times higher, and the values of amylin and PP were 1.1 times higher. MCP-1. The PAI-1 index was significantly lower in the group with decreased GFR (1.3 times lower compared to the normal renal function group) (Table 3).

In individuals with renal dysfunction and DLP, the level of lipocalin-2 was 1.4 times lower, and amylin was 1.1 times higher than in individuals without DLP.

In patients with normal renal function, significant differences between the groups with and without DLP were obtained for more adipocytokines compared to individuals with renal dysfunction. In individuals with GFR ≥ 90 mL/min/1.73 m^2^ with DLP, higher values of resistin, amylin, C-peptide, leptin, and lower levels of adiponectin were recorded compared to patients without DLP (Table 4).

In individuals with DLP, weak negative associations were obtained between GFR and amylin, ghrelin, GIP, PP, TNF-α, adipsin, lipocalin-2, and C-peptide. The links between GFR and leptin and resistin levels were close to moderate. We also revealed weak direct associations between GFR and PAI-1 (Figure 4). 

When the level of lipocalin-2 was raised by 1 mcg/mL in DLP patients, the likelihood of having renal impairment rose by 0.1%. The likelihood of having a reduced GFR increased by 0.2% for every 1 pg/mL rise in GLP-1, 0.3% for every 1 mcg/mL increase in resistin, 0.4% for a 1 pg/mL increase in GIP, 0.6% for a 1 pg/mL increase in PYY, 3.8% for a 1 mg/mL increase in IL-6, and 0.4% for a 1 pg/mL increase in PP. An increase in TNFa levels by 1 pg/mL in individuals with DLP considerably increases the likelihood of having a GFR of less than 90 mL/min/1.73 m^3^ (by 11.2%). A 1 ng/mL rise in PAI-1 was linked to a 2.1% reduction in the likelihood of having a lower GFR (Table 5).

When all of the adipokines linked to impaired renal function in people with DLP were included in the multivariate regression analysis, it was discovered that an increase in resistin of 1 mcg/mL was linked to a 0.2% greater likelihood of having renal dysfunction and a 1.1% increase in the likelihood of experiencing renal impairment was linked to a rise in GIP levels (Table 6).

## 4. Discussion

Atherogenic dyslipidemia is a risk factor for the development of chronic kidney disease (CKD) [15]. Some studies have studied the relationship between dyslipidemia and the occurrence of CKD in middle-aged people with a high risk of cardiovascular diseases [15,16]. Even after adjusting for confounding variables, a higher quartile of the TG: HDL-C ratio at baseline was substantially linked to a larger reduction in eGFR and a rise in urine protein excretion during the 2-year study period [16]. An increased level of LDL-C is associated with the development of CKD and a decrease in eGFR in young and middle-aged working men without hypertension and/or diabetes [17]. Oxidized LDL-C contributes to the development of glomerulosclerosis and arteriosclerosis due to the infiltration of monocytes or macrophages and the overexpression of adhesion molecules [18], which leads to ischemic damage to the renal glomeruli.

GIP is an incretin hormone secreted by intestinal neuroendocrine cells during food intake, which increases insulin secretion [19]. GIP, similar to GLP-1, reduces appetite, contributing to weight loss, which in turn improves metabolic control and is the basis for prescribing GLP1RA, such as liraglutide and semaglutide. Moreover, GIP reduces the secretion of gastric acid and promotes the secretion of glucagon, which is an advantage when prescribing combined preparations of GIP receptor agonists and GLP-1 (tirzepatide) for overweight and obese patients. The agonism of GIP receptors in adipocytes increases blood flow in adipose tissue and promotes lipid uptake by adipose tissue, thus regulating postprandial lipid clearance and, potentially, overall lipid homeostasis [20].

The circulating level of endogenous GIP depends on renal metabolism and is elevated in people with chronic kidney disease [21,22]. The effect of GIP on the kidneys has not been thoroughly studied due to the absence of detectable renal expression of the GIP receptor (GIPR) [23]. GIPR is expressed in cardiomyocytes in the heart and in some endothelial cells of blood vessels. GIP may play an indirect role in the development of atherosclerosis through the regulation of inflammation caused by macrophages, the formation of foam cells, the proliferation of SMC, and the remodeling of arteries [24]. Atherosclerotic lesions of renal atherias contribute to a decrease in GFR [25], so we can assume that GIP is indirectly associated with a decrease in renal function in patients with DLP.

Resistin is a protein that belongs to the family of polypeptides produced primarily by leukocytes and mononuclear cells associated with adipose tissue [26]. Accumulating data confirm that resistin promotes endothelial dysfunction and pro-inflammatory activation, which leads to an acceleration of the development of subclinical atherosclerosis [27,28,29,30,31]. Recently discovered associations of resistin with carotid artery atherosclerosis [32] and markers of chronic renal failure in patients with different levels of GFR and metabolic syndrome confirm the active role of resistin in the manifestation of diffuse atherosclerotic damage [33]. Resistin increases the expression of endothelin, adhesion molecules, and matrix metalloproteinases, contributing to systemic vascular dysfunction and adversely affecting eGFR [26,30,31,32,33,34]. A decrease in GFR may be a consequence of a resistin-mediated vascular disorder, even in the early stages of atherosclerotic disease. From another point of view, since the kidneys play an important role in the catabolism of small polypeptides, such as resistin, a violation of the functional renal parenchyma can increase the concentration of resistin [35,36]. A growing body of data confirms the relationship between reduced GFR and low-intensity inflammation [33,35] and atherosclerosis [37,38]. The K. Dimitriadis study revealed an independent association of resistin with eGFR in young people with arterial hypertension, which suggests the involvement of resistin in the progression of kidney damage [39].

As a result, we may presume that resistin and GIP are linked to a decline in eGFR in people with atherogenic DLP, which could serve as a foundation for more extensive research into these biomarkers. Large prospective studies are needed to establish causal relationships between dyslipidemia, adipokines, and CKD.

The limitations of the study include the assessment of kidney function only by a single measurement of creatinine and calculation of GFR and the lack of data on changes in urinary sediment.

## 5. Conclusions

Resistin and GIP, regardless of the presence of abdominal obesity, are associated with the presence of renal dysfunction in young people with dyslipidemia.

## Figures and Tables

**Figure 1 jpm-13-01238-f001:**
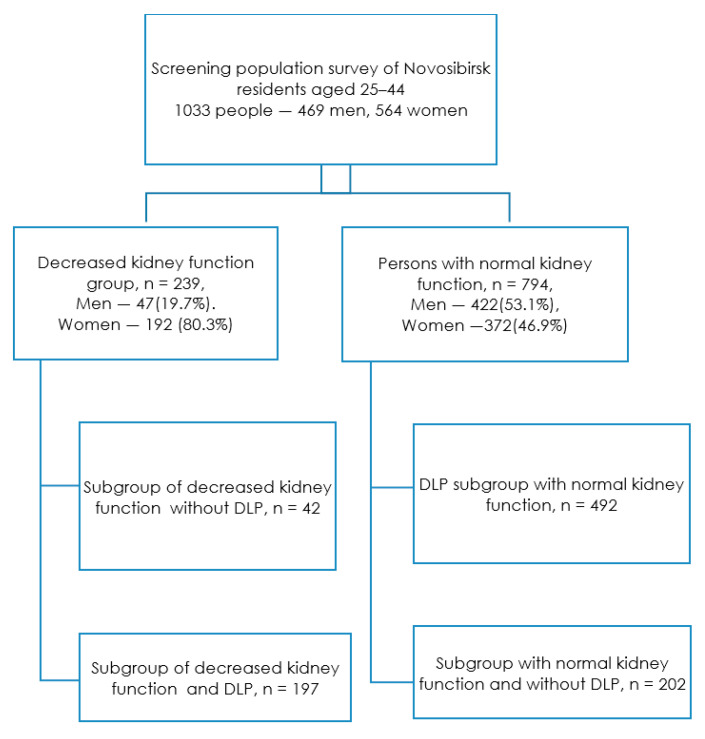
Study design. Footnote: DLP—dyslipidemia, decreased kidney function—GFR < 90 mL/min/1.73 m^2^.

**Figure 2 jpm-13-01238-f002:**
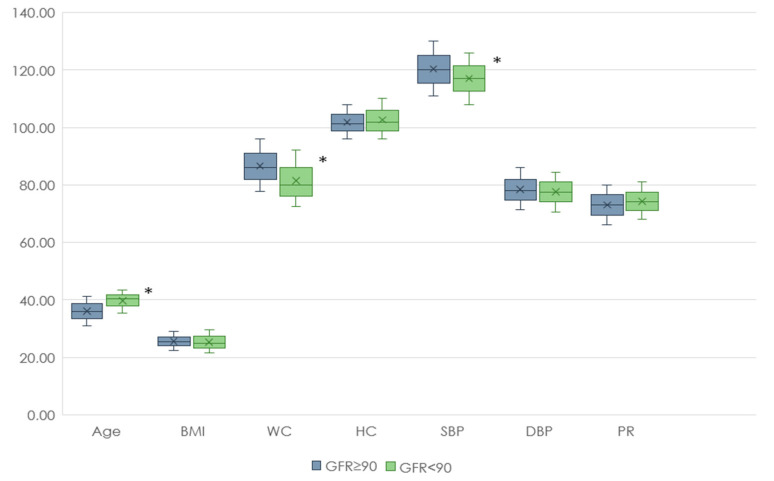
Anthropometric data of patients with renal dysfunction and normal renal function. Footnote: BMI—body mass index, WC—waist circumference, HC—hip circumference, SBP—systolic blood pressure, DBP—diastolic blood pressure, PR—pulse rate, *—*p* < 0.05.

**Figure 3 jpm-13-01238-f003:**
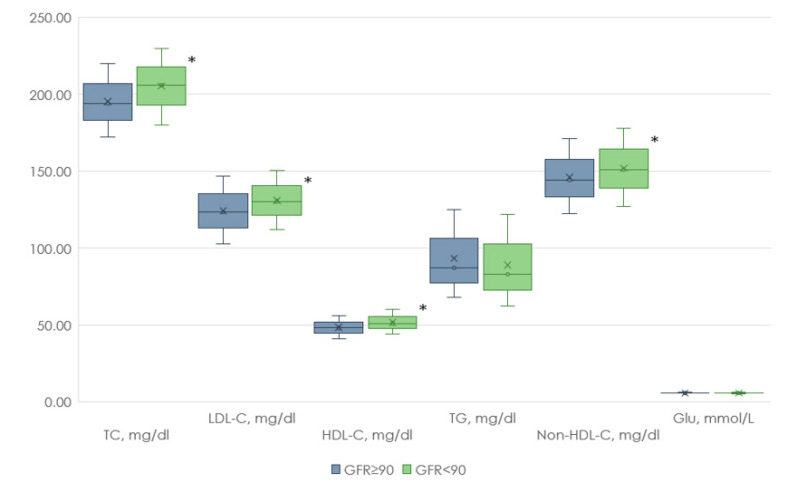
Biochemical parameters of patients with renal dysfunction and normal renal function. Footnote: TC—total cholesterol, LDL-C—low-density lipoprotein cholesterol, HDL-C—high-density lipoprotein cholesterol, TG—triglycerides, Non-HDL-C—Non-HDL cholesterol, Glu—glucose, *—*p* < 0.05.

**Figure 4 jpm-13-01238-f004:**
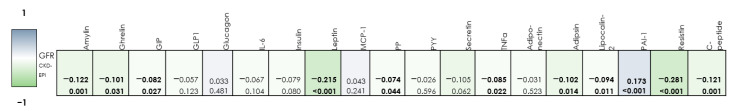
Correlation matrix of the relationship between the studied adipocytokines and the level of GFR in individuals with DLP. (bold—significant correlations)

**Table 1 jpm-13-01238-t001:** Quantitative indicators of risk factors of reduced GFR in subgroups.

Subgroups Indicator	GFR ≥ 90/DLP (-). n = 202		GFR < 90/DLP (-). n = 42		*p*	GFR ≥ 90/DLP (+). n = 592		GFR < 90/DLP (+). n = 197		*p*
	Me	[25%; 75%]	Me	[25%; 75%]		Me	[25%; 75%]	Me	[25%; 75%]	
Age, years	33.41	29.33; 38.63	37.88	30.17; 42.52		37.00	31.52; 41.67	40.67	35.88; 43.38	
BMI, kg/m^2^	23.78	21.10; 27.08	22.29	19.50; 25.07	0.040	25.89	22.95; 29.50	25.73	21.95; 30.10	0.368
WC, cm	82.50	74.00; 90; 00	73.90	67.30; 78.10	<0.001	88.00	79.00; 98.00	84.00	74.65; 94.00	<0.001
SBP, mmHg.	117.50	108.75; 127.13	113.50	104.63; 120.75	0.011	121.00	111.00; 131.50	117.50	109.00; 127.50	0.027
DBP, mmHg.	76.50	70.88; 83.63	72.75	68.88; 79.75	0.048	79.50	72.00; 87.00	77.50	71.50; 85.75	0.522
Heart rate, beats/min	70.00	65.00; 79.00	74.00	66.00; 80.50	0.301	74.00	66.00; 81.00	74.00	68.00; 81.00	0.818
Albumin, mmol/l	42.70	40.80; 44.33	41.65	39.10; 43.83	0.022	42.90	40.90; 47.20	42.20	40.60; 44.00	0.018
LDL-C, mg/dL (mmol/L)	97.30 (2.52)	86.20 (2.23); 109.40 (2.83)	99.10 (2.56)	88.00 (2.28); 107.45 (2.78)	0.550	133.00 (3.44)	119.40 (3.09); 154.95 (4.00)	137.20 (3.55)	125.00 (3.23); 159.10 (4.11)	0.060
Non-HDL-C, mg/dL (mmol/L)	112.00 (2.9)	102.75 (2.66); 123.00 (3.18)	112.50 (2.91)	98.75 (2.55); 123.25 (3.19)	0.875	155.00 (4.01)	138.00 (3.60); 180.00 (4.66)	156.00 (4.03)	142.00 (3.67); 183.50 (4.74)	0.288
HDL-C, mg/dL (mmol/L)	53.00 (1.37)	48.00 (1.24); 61.00 (1.58)	58.00 (1.50)	53.75 (1.39); 65.00 (1.68)	0.002	46.95 (1.21)	40.00 (1.03); 54.00 (1.40)	49.00 (1.27)	42.00 (1.09); 58.50 (1.51)	0.001
TG, mg/dL (mmol/L)	68.00 (0.77)	53.75 (0.61); 92.00 (1.04)	61.50 (0.69)	48.25 (0.55); 82.25 (0.93)	0.076	94.00 (1.06)	67.00 (0.76); 143.95 (1.63)	89.00 (1.00)	64.00 (0.72); 126.00 (1.42)	0.251
Glucose, mmol/L	5.73	5.39; 6.04	5.62	5.10; 6.04	0.373	5.83	5.41; 6.20	5.73	5.41; 6.04	0.062

Note: GFR—glomerular filtration rate, SBP—systolic blood pressure, DBP—diastolic blood pressure, LDL-C—low-density lipoproteins cholesterol, HDL–C—high-density lipoproteins cholesterol, Non-HDL-C—cholesterol—non–high–density lipoproteins, TG—triglycerides, BMI—body mass index, WC—waist circumference, HC—hip circumference, Me—median, 25% and 75%—1st and 3rd quartiles.

**Table 2 jpm-13-01238-t002:** The presence of risk factors in persons with GFR ˂ 90 mL/min/1.73 m^3^, depending on the presence of DLP.

Indicator		No DLP (n = 42)	DLP (n = 197)	*p*
Gender (men)		3 (7.1%)	44 (22.3%)	0.025
Smoking	Both sexes	15 (35.7%)	44 (22.4%)	0.071
	Men	1 (33.3%)	17 (38.6%)	0.855
	Women	14 (35.9%)	27 (17.8%)	0.014
Blood pressure ≥ 140/≥90 mmHg.	Both sexes	1 (2.4%)	37 (18.8%)	0.008
	Men	0	19 (43.2%)	0.140
	Women	1 (2.6%)	18 (11.8%)	0.086
WC ≥ 80 cm in women and ≥94 cm in men, cm	Both sexes	7 (16.7%)	89 (45.2%)	0.001
	Men	1 (33.3%)	18 (40.9%)	0.796
	Women	6 (15.4%)	71 (46.4%)	<0.001
Glucose ≥ 6.1 mmol/L	Both sexes	9 (21.4%)	47 (23.9%)	0.736
	Men	2 (66.7%)	20 (45.5%)	0.476
	Women	7 (17.9%)	27 (17.6%)	0.965
Alcohol consumption ≥ 14/day or ≥4 on one day for men; ≥7/day or ≥3 on one day for women	Both sexes	4/34 (11.8%)	12/138 (9.3%)	0.525
	Men	0 (0%)	9/36 (25%)	0.999
	Women	4/31 (12.9%)	3/102 (2.9%)	0.051

Note: GFR—glomerular filtration rate, BP—blood pressure, AO—abdominal obesity, DLP—dyslipidemia.

**Table 3 jpm-13-01238-t003:** Levels of human adipokines (Me [25%; 75%]).

Adipokines	Group 1 GFR ≥ 90 mL/min/1.73 m^2^, n = 794	Group 2 GFR < 90 mL/min/1.73 m^2^, n = 239	*p*
Adiponectin, mcg/mL	42.2 [28.1; 130.5]	37.8 [32.2; 136.0]	0.574
Adipsin, mcg/mL	11.4 [8.8; 14.5]	13.4 [11.5; 14.6]	0.0001
Lipocalin-2, pcg/mL	4.3 [2.2; 10.6]	10.0 [3.5; 13.3]	0.0001
Resistin, ng/mL	73.1 [23.6; 555.2]	560.6 [397.2; 689.0]	0.0001
Amylin, pg/mL	5.8 [0.7; 14.1]	5.9 [5.4; 13.6]	0.0001
IL-6, pg/mL	1.1 [0.6; 2.2]	1.5 [0.8; 5.1]	0.004
PAI-1, ng/mL	25.2 [17.5; 37.7]	19.5 [11.0; 30.0]	0.001
C-peptide, ng/mL	0.7 [0.3; 1.2]	1.0 [0.7; 1.4]	0.0001
Insulin, pmol/L	458.2 [296.0; 675.4]	509.5 [398.6; 700.6]	0.038
Leptin, ng/mL	4.1 [1.5; 7.8]	6.4 [3.5; 10.8]	0.0001
MCP-1, pg/mL	237.2 [152.9; 318.7]	241.8 [178.5; 321.1]	0.497
Ghrelin, pg/mL	30.7 [18.4; 83.1]	36.9 [29.2; 88.4]	0.009
TNFa, pg/mL	4.6 [3.0; 6.9]	6.6 [3.4; 8.8]	0.0001
GIP, pg/mL	23.9 [14.9; 46.5]	29.2 [19.0; 57.2]	0.0001
Glucagon, pg/mL	11.7 [7.2; 25.0]	11.8 [6.8; 19.2]	0.661
PP, pg/mL	38.4 [20.0; 73.8]	43.3 [29.5; 84.5]	0.0001
GLP1, pg/mL	252.3 [162.5; 451.8]	396.0 [201.2; 547.3]	0.0001
PYY, pg/mL	52.4 [36.1; 71.2]	61.4 [40.8; 88.1]	0.015
Secretin, pg/mL	21.5 [14.3; 59.9]	31.7 [19.4; 87.8]	0.001

Note: PAI-1—plasminogen activator inhibitor type 1, IL–6—interleukin 6, MCP-1—monocytic chemoatractant protein–1, TNFa—tumor necrosis factor-alpha, GIP—glucose-dependent insulinotropic polypeptide, PP—pancreatic polypeptide, GLP1—glucagon-like peptide-1, PYY—peptide tyrosine-tyrosine, GFR—glomerular filtration rate, Me—median, 25% and 75%—1st and 3rd quartiles.

**Table 4 jpm-13-01238-t004:** Levels of human adipokines studied with and without AO (Me [25%; 75%])**.**

Adipokines	GFR ≥ 90 mL/min/1.73 m^2^ (n = 794)		*p*	GFR < 90 mL/min/1.73 m^2^ (n = 239)		*p*
	Without DLP (n = 202)	With DLP (n = 492)		Without DLP (n = 42)	With DLP (n = 197)	
Adiponectin, mcg/mL	64.8 [35.0; 150.2]	40.5 [26.8; 120.7]	0.005	115.1 [34.4; 182.2]	37.4 [31.4; 131.3]	0.110
Adipsin, mcg/mL	11.6 [9.0; 14.8]	11.4 [8.7; 14.3]	0.491	13.2 [12.2; 14.9]	13.4 [11.0; 14.6]	0.656
Lipocalin-2, pcg/mL	438.3 [192.2; 1030.4]	429.4 [224.6; 1077.0]	0.303	1210.0 [504.7; 1725.3]	845.2 [319.9; 1294.4]	0.005
Resistin, mcg/mL	39.7 [20.5; 538.6]	92.2 [25.2; 559.2]	0.015	580.0 [314.2; 676.4]	559.2 [404.7; 690.1]	0.952
Amylin, pg/mL	5.5 [0.7; 9.9]	5.9 [0.8; 14.2]	0.005	5.7 [5.3; 6.5]	6.03 [5.5; 14.0]	0.037
IL-6, pg/mL	1.1 [0.4; 2.0]	1.2 [0.6; 2.3]	0.10956	1.4 [0.7; 2.8]	1.5 [0.9; 5.1]	0.544
PAI-1, ng/mL	23.8 [16.5; 34.0]	26.0 [17.6; 39.0]	0.117	20.2 [11.8; 30.2]	19.5 [10.8; 30.0]	0.932
C-peptide, ng/mL	0.5 [0.2; 0.9]	0.8 [0.3; 1.3]	0.0001	1.0 [0.7; 1.2]	1.0 [0.6; 1.4]	0.966
Insulin, pmol/L	497.0 [323.0; 614.1]	446.0 [296.0; 702.9]	0.917	505.7 [390.4; 646.6]	546.5 [398.6; 719.0]	0.581
Leptin, ng/mL	2487.4 [1186.4; 5552.8]	4650.9 [1780.0; 8961.1]	0.0001	5227.2 [2183.3; 6880.7]	6569.9 [3528.9; 15240.1]	0.060
MCP-1, pg/mL	234.6 [163.7; 317.6]	240.2 [149.7; 319.4]	0.855	267.9 [200.7; 338.8]	239.8 [161.4; 319.8]	0.264
Ghrelin, pg/mL	29.2 [13.5; 54.7]	30.8 [18.6; 90.6]	0.068	33.8 [19.6; 87.5]	37.0 [29.2; 88.4]	0.352
TNFα, pg/mL	4.2 [2.7; 6.6]	4.65 [3.1; 6.9]	0.119	6.9 [3.0; 8.8]	6.5 [3.5; 8.8]	0.890
GIP, pg/mL	22.7 [13.8; 45.0]	24.8 [15.5; 46.7]	0.136	27.6 [18.3; 51.5]	29.9 [19.7; 59.4]	0.269
Glucagon, pg/mL	11.0 [5.5; 25.6]	11.9 [7.7; 24.5]	0.324	11.2 [5.4; 18.7]	12.0 [9.0; 19.2]	0.310
PP, pg/mL	36.4 [18.7; 66.0]	40.1 [21.3; 75.7]	0.066	66.9 [34.7; 84.4]	41.6 [28.9; 84.9]	0.225
GLP1, pg/mL	232.5 [144.0; 447.6]	263.5 [168.8; 453.4]	0.094	244.8 [209.0; 463.6]	409.0 [199.9; 556.0]	0.325
PYY, pg/mL	53.7 [41.0; 73.8]	51.3 [36.0; 72.1]	0.572	65.2 [49.2; 121.5]	61.3 [37.8; 80.7]	0.111
Secretin, pg/mL	25.7 [17.2; 50.7]	20.5 [13.0; 62.7]	0.381	48.0 [27.6; 116.1]	30.3 [19.1; 76.9]	0.082

Note: PAI-1—plasminogen activator inhibitor type 1, IL–6—interleukin 6, MCP-1—monocytic chemoatractant protein–1, TNFa—tumor necrosis factor-alpha, GIP—glucose-dependent insulinotropic polypeptide, PP—pancreatic polypeptide, GLP1—glucagon-like peptide-1, PYY—peptide tyrosine-tyrosine, GFR—glomerular filtration rate, Me—median, 25% and 75%—1st and 3rd quartiles.

**Table 5 jpm-13-01238-t005:** Results of a logistic regression analysis of the association of adipokines with the risk of having a GFR ˂ 90 mL/min/1.73 m^2^ in young persons with DLP with standardization by gender, age, and smoking status.

Indicators	Logistic Regression Analysis			
	OR	95% Confidence Interval (CI)		*p*
		Lower Limit	Upper Limit	
C-peptide, ng/mL	1.175	0.936	1.475	0.163
GIP, пг/мл	1.004	1.001	1.008	0.033
GLP1, пг/мл	1.002	1.001	1.003	0.0001
PAI-1, нг/мл	0.979	0.960	0.998	0.032
IL-6, пг/мл	1.038	1.002	1.075	0.037
Insulin, ng/mL	1.000	1.000	1.000	0.705
Adipsin, mcg/mL	1.000	0.988	1.013	0.946
Lipocalin-2, mcg/mL	1.001	1.001	1.002	0.001
Amylin, pg/mL	1.006	0.988	1.025	0.489
Ghrelin, pg/mL	1.0	0.999	1.001	0.919
Leptin, ng/mL	1.000	1.001	1.000	0.915
MCP-1, pg/mL	1.000	0.999	1.002	0.781
PP, pg/mL	1.004	1.001	1.008	0.026
PYY, pg/mL	1.006	1.001	1.011	0.038
Secretin, pg/mL	1.000	1.001	1.000	0.314
TNF-a, pg/mL	1.112	1.055	1.171	0.0001
Adiponectin, mcg/mL	1.000	0.997	1.002	0.640
Resistin, mcg/mL	1.003	1.002	1.004	0.0001

Note: PAI-1—type 1 plasminogen activator inhibitor. IL-6—interleukin 6. MCP-1—monocytic chemoatractant protein-1. TNF-a—tumor necrosis factor-alpha. GIP—glucose-dependent insulinotropic polypeptide, PP—pancreatic polypeptide, GLP1—glucagon-like peptide-1. PYY—tyrosine-tyrosine peptide. GFR—glomerular filtration rate. DLP—dyslipidemia. OR—odds ratio.

**Table 6 jpm-13-01238-t006:** Multivariate regression analysis of the association of adipokines with the chance of having a GFR ˂ 90 mL/min/1.73 m^2^ against the background of DLP with standardization by gender, age, smoking status, and the presence of AO.

Indicators	Logistic Regression Analysis			
	OR	95% Confidence Interval (CI)		*p*
		Lower Limit	Upper Limit	
GIP, pg/mL	1.011	1.001	1.021	0.042
GLP-1, pg/mL	1.001	1.000	1.003	0.093
Lipocalin-2, mcg/mL	1.000	0.999	1.000	0.239
PP, pg/mL	0.999	0.991	1.006	0.714
TNF-a, pg/mL	1.043	0.950	1.145	0.382
Resistin, mcg/mL	1.002	1.001	1.004	0.008
PYY, pg/mL	0.998	0.990	1.006	0.583
IL-6, пг/мл	0.979	0.914	1.048	0.542
C-peptide, ng/mL	1.169	0.767	1.782	0.467

Note: IL-6—interleukin 6. TNF-a—tumor necrosis factor-alpha. GIP—glucose-dependent insulinotropic polypeptide, PP—pancreatic polypeptide, GLP1—glucagon-like peptide-1. PYY—tyrosine-tyrosine peptide. GFR—glomerular filtration rate. DLP—dyslipidemia. AO—abdominal obesity. OR—odds ratio.

## Data Availability

Not applicable.

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
