# Peer review of "Associations of Adipocytokines and Early Renal Dysfunction in Young People on the Background of Dyslipidemia"

_jpm, 2023, doi:10.3390/jpm13081238_

Round 1

Reviewer 1 Report

The JPM instructions for authors state that:

  • SI Units (International System of Units) should be used. Imperial, US customary and other units should be converted to SI units whenever possible.

Cholesterol should therefore be reported in mmol/L, not the  archaic units mg/dL. Other assays will need conversion to correct units if appropriate.

In the methods section it is noted that alcohol consumption was assessed in the study but alcohol is then not considered anywhere else in the manuscript. Alcohol consumption is known to be associated with CKD, LDL, and adipocytokines.

Some example papers:

https://www.ncbi.nlm.nih.gov/pmc/articles/PMC6767945/

https://www.ncbi.nlm.nih.gov/pmc/articles/PMC7230699/

https://diabetesjournals.org/care/article/27/1/184/26655/Effect-of-Moderate-Alcohol-Consumption-on

Therefore, this is an important variable that must be considered in any analysis. It would be perfectly plausible that all of the differences in the groups could be accounted for by their alcohol consumption, with no association shown in the current manuscript surviving corrections. Given that studies of mortality in Russia have had to define 'low alcohol consumers' as those who drink less than 1 bottle of vodka / week [because there were too few people in the low group to analyse if non-drinkers were specified  ( https://www.thelancet.com/journals/lancet/article/PIIS0140-6736(13)62247-3/fulltext#seccestitle80  )], this does suggest that as this manuscript reports a study that was carried out in a Russian population the consumption of alcohol is likely to be very significant.

At this point therefore, although the study has some potentially interesting findings, it needs to be taken back and associations with alcohol need to be corrected for before it can be resubmitted.

It would also be reasonable to consider whether corrections for obesity, insulin resistance, etc affect interpretation of the data.

Author Response

Dear reviewer, We thank you for your interest in our work!

  • We have entered data in mmol/L in the tables
  • We added all information about alcohol consumption into the text of the article, but there was no difference between groups with low GFR and DLP and without DLP, so we didn’t include it in the logistic regression model.
  • We made corrections for abdominal obesity at every stage of analysis.

Reviewer 2 Report

Manuscript ID: jpm-2518697 The study offers insights into the relationship between lipid profile (Dyslipidemia) and CKD.    Below are my suggestions:   Figure 1: The letters/words are not clear and correct the footnotes- spelling errors   How does the sample size of your study impact the statistical reliability of the findings?   Discussion can have some analyzation of results of other studies in this line   Present the study's conclusion in a clear manner    What are  the possible implications of the study's findings for early prediction of renal impairment   What are the limitations of your study?   What are the long term impact of lipid profile parameters on the progression of CKD in young people with dislipidemia?   In what ways could the cross- sectional design and sample population impact the generalization of the study's conclusions to other ethnic groups or people with different stages of CKD   What further research would be beneficial to establish causality?    

The whole manuscript needs language correction

Author Response

Dear reviewer, We thank you for your interest in our work!

- We tried to make Figure 1 clearer and corrected the footnotes.

- The sample size was calculated using the formula: According to calculations, the required sample size was 288 people. We have added this information to the text. The sample size is sufficient to evaluate the results.

- The limitations of the study are described at the end of the discussion.

- If these indicators are detected in the early stages of CKD, then in the future they can be used for early prognosis of renal failure but large prospective studies are needed.

- The long-term impact of lipid profile parameters on the progression of CKD in young people is described in the first paragraph: “Atherogenic dyslipidemia is a risk factor for the development of chronic kidney dis-ease (CKD) [15]. Some studies have studied the relationship between dyslipidemia and the occurrence of CKD in middle-aged people with a high risk of cardiovascular diseases [15, 16]. Even after adjusting for confounding variables, higher quartile of TG: HDL-C ratio at baseline was substantially linked with larger reduction in eGFR and rise in urine protein excretion during the 2-year study period [16]. An increased level of LDL-C is associated with the development of CKD and a decrease in eGFR in young and middle-aged working men without hypertension and/or diabetes [17]. Oxidized LDL-C contributes to the development of glomerulosclerosis and arteriosclerosis due to the infiltration of monocytes or macrophages and the overexpression of adhesion molecules [18], which leads to ischemic damage to the renal glomeruli.”

- Population studies reflect the state of the problem at one point in time in a particular region (in particular in the Novosibirsk region of Russia). The data can only be extrapolated to populations living in similar climate-geographic conditions. Large prospective studies are needed to establish causal relationships.

Round 2

Reviewer 1 Report

Much improved. I can understand why alcohol consumption was left out of the original manuscript [because it was not relevant], but given external views [prejudices] about Russian society, it is necessary to demonstrate that it was not relevant! The stats do show that.

There is clearly an association between some of the cytokines tested and renal dysfunction. There are at least 2 possible ways of interpreting this: Higher levels of the cytokine cause dysfunction; OR, renal dysfunction causes higher levels of the cytokine. In the case of GIP, it is known that the kidney is the main route of removal. Therefore, it is possible that GIP is higher because of renal dysfunction, but equally that renal dysfunction is worsened by GIP - Correlation does not prove causation... That has been noted in the discussion, and similar comments have been made about resistin. 

Overall, the changes made have satisfied my original comments.

Therefore